# Driving Factors Influencing Soil Microbial Community Succession of Coal Mining Subsidence Areas during Natural Recovery in Inner Mongolia Grasslands

**DOI:** 10.3390/microorganisms12010087

**Published:** 2023-12-31

**Authors:** Dongqiang Lu, Zhen Mao, Yan Tang, Bo Feng, Liang Xu

**Affiliations:** School of Environment Science and Spatial Informatics, China University of Mining and Technology, Xuzhou 221116, China; ludongqiang@cumt.edu.cn (D.L.); tangyan@cumt.edu.cn (Y.T.); fengbo@cumt.edu.cn (B.F.); xuliang0911@cumt.edu.cn (L.X.)

**Keywords:** coal mining subsidence areas, driving factors, grassland, microbial community diversity, natural recovery

## Abstract

Soil microorganisms significantly influence the energy flow and material cycle of soil ecosystems, making them highly susceptible to environmental changes, such as those induced by mining activities. Studying the succession of soil microbial communities after mining subsidence is crucial for comprehending the significance of soil microbes in the natural recovery process following subsidence. Therefore, the soil properties, vegetation communities, and soil microbial communities of the subsidence area, as well as unexploited areas, were analyzed during the natural restoration process (1, 2, 5, 10, and 15 years). The results demonstrate that mining subsidence has a significant impact on the aboveground vegetation community, soil properties, and microbiological community. Following an extended period of natural recovery, a new stable state has emerged, which differs from that observed in non-subsidence areas. The total nitrogen, nitrate nitrogen, and ammonium nitrogen amounts may be key factors driving the natural recovery of bacterial communities, and total potassium and available potassium may be key factors driving the natural recovery of fungal communities. The natural recovery mechanism of soil microorganisms was analyzed along with the changes related to vegetation and soil physicochemical properties. The mechanism was explained from three perspectives, namely, plant-led, soil-led, and soil-microbial-led, which could provide a theoretical basis for the natural restoration of grassland ecosystems and provide guidance for the treatment of coal mining subsidence areas.

## 1. Introduction

As one of the most important material energy sources, coal accounts for about 74% of China’s primary energy consumption. The rapid development of the coal mining industry plays a vital role in promoting the development of human society and civilization [1,2]. As the world’s largest coal producer, statistically, China mines about 4.56 billion tons of coal annually, accounting for more than half of the world’s total output [3]. A total of 96% of China’s coal mining is underground mining, and 4% is open-pit mining. However, coal hollowing causes serious disturbances to the ecosystem surrounding the mining area, and environmental factors change accordingly; these disturbances introduce a series of geological disasters and environmental problems, such as surface subsidence, ground fissures, ecological landscape degradation, and groundwater pollution [4,5,6]. As the demand for minerals and energy continues and as mining activities continue to evolve, coal mining subsidence areas continue to increase at a rapid rate [7,8,9].

With the emergence and development of high-volume sequencing technology, the role of the soil microbial community in mine remediation has gradually attracted attention [10]. By comparing the vegetation characteristics, microbial diversity, and microbial community structure in the coal mining subsidence area with the unexploited area, the recovery status of the coal mining subsidence area in the mining area can be evaluated [11,12]. At present, there have been many studies on the ecological recovery of coal mining subsidence sites around the world. Some studies have shown that with the restoration of vegetation in the collapse area, the community structure and function of soil bacteria will change with changes in vegetation; the physical and chemical properties, as well as the microbial community diversity of soil, are positively correlated with soil nutrients but negatively correlated with soil physical properties and soil pH values at various developmental stages [13,14]. Relevant studies have shown that the organic matter and nutrients in soil from disturbed areas have been greatly reduced [15]. The assessment of the microbial community structure in coal mining subsidence areas shows that when the microbial community structure is damaged, the relevant material cycle of the soil and the sustainability of the ecosystem will be affected to varying degrees [16]. Disturbances cause a decline in soil microbial diversity and affect the activity of related microorganisms, thereby affecting energy flow and material cycle and ultimately reducing soil nutrients [17].

Studies have shown that after 5 years of vegetation reconstruction, the diversity of bacterial communities in artificial and natural restoration areas does not reach pre-subsidence index values. However, after 15 years of vegetation reconstruction, the diversity of bacterial communities can reach pre-subsidence levels, and compared with natural restoration, artificial remediation can accelerate the restoration of soil variables [18]. Different research has demonstrated variations in microbial diversity, abundance, and functions between soils that have been reclaimed and those that have undergone natural restoration in subsidence areas [19,20,21]. However, over time, microbial diversity and ecosystem sustainability can be restored to be as close to the pre-disturbance state as possible through artificial restoration [22]. However, most of these studies focus on artificial restoration in collapse areas and less on the differences in soil microbial diversity in coal mining collapse areas under natural recovery conditions across different recovery years. Grassland ecosystems have remarkable resilience; studying the natural restoration of grassland vegetation and soil microorganisms after subsidence is significant in guiding artificial restoration.

Therefore, this project focuses on a grassland mining area. Soil samples with different subsidence years (1, 2, 5, 10, and 15 years) that have not been artificially restored and unexploited areas were selected as the research objects. This research mainly focuses on two aspects: (1) investigating the mechanism of the vegetation community, soil properties, and the soil microbial community during natural recovery after subsidence and (2) examining the main driving factors of the natural recovery mechanism in the soil microbial community.

## 2. Materials and Methods

### 2.1. Study Area

The grassland mining area selected by this institute is the Zhalainuoer mining area, located in the west of Hulunbuir City, Inner Mongolia Autonomous Region, administratively under the jurisdiction of Hulunbuir City and Manzhouli City. The mining area is about 23.8 km long from north to south, 22.83 km wide from east to west, and covers an area of 543.43 km^2^. The geographical coordinates are 49°19′~49°46′ north latitude and 117°12′~117°53′ east longitude (Figure 1).

This area belongs to the moderate temperate semi-arid continental monsoon climate, with cold winters and hot summers, temperature changes, a freezing period of up to 7 months, and a summer of only about 5 months. According to Manzhouli City and Zhaqu meteorological station data, the lowest temperature is −42.7 °C, the highest temperature is +37.8 °C, and the annual average temperature is generally 0.2 to −2 °C; the evaporation is the largest from April to September, and the annual evaporation is generally 1200~1500 mm. The southwest wind is the most common throughout the whole year, followed by the northwest wind in June, July, and August. The wind speed is generally 2–5 m/s, and the maximum wind speed is 20 m/s. The wind is the lowest in January, February, and December. The wind is the strongest in March, April, May, and November [23].

### 2.2. Research Methods

In July 2022, under the leadership and guidance of staff familiar with the situation of the mining area, areas with five subsidence years of 1, 2, 5, 10, and 15 years without artificial restoration and one unexploited area were selected as the sampling group.

Before sampling, the site was investigated, and related data were collected. The sampling range was divided into sampling units according to soil type, topography, and other factors, and the soil of each sampling unit was kept as consistent as possible. According to the size of the subsidence plot on site, we set the number of sampling points to 6 to ensure the greatest possible representativeness of the sampling. At the same time, sampling was carried out according to the principle of “multi-point mixing”. All samples were taken in duplicate for further analysis. One fresh sample was stored cryogenically in sterile centrifuge tubes to determine soil microorganisms, and the other was used to determine soil physicochemical properties. The removed surface plants were weighed fresh in sterile bags, then taken back to the laboratory to dry to a constant weight, and their dry weight was measured.

### 2.3. Sample Collection and Processing Methods

#### 2.3.1. Plant Samples

To clarify the effect of plant community development on soil properties and bacterial community changes, two duplicate quadrats (1 × 1 m grassland communities) were randomly selected in each plot to investigate the component and diversity of the plant community. We recorded all herb species, names, quantity, coverage, and maximum/average height in each quadrat (Appendix A). The plant coverage was estimated visually by observers. A single plant was used to calculate the number of individuals of each species in each plot, and the plant density of each plot was calculated. Additionally, aboveground portions of all vegetation were cut and dried at 70 °C for 48 h to obtain the aboveground biomass. The species Pielou uniformity (E) and the Shannon–Wiener index (H) of the plant communities were calculated as a characterization of plant community diversity.

#### 2.3.2. Soil Samples

Soil moisture (SW) and pH of samples were measured simultaneously on-site using the soil temperature, moisture, salt, and pH velocity tester (model: HM-WSYP). After removing the topsoil, the soil sample was collected with a depth of 0~10 cm, and the mixed soil sample was made for subsequent analysis. The soil sample was sealed with a ziplock bag and returned to the laboratory for pretreatment to determine physical and chemical properties. Soil organic matter (SOM) was detected by potassium dichromate oxidation–external heating. The semi-trace Kjeldahl method was used to detect total nitrogen (TN) in soil. Total phosphorus (TP) and total potassium (TK) were tested separately using the sodium hydroxide melt–molybdenum antimony anti-colorimetric method and sodium hydroxide molten flame photometry [24]. Alkali-hydrolyzable nitrogen (AN), available phosphorus (AP), and available potassium (AK) were determined according to the methods of Olsen [25] and Colwell [26], respectively.

#### 2.3.3. Soil Microbial Samples


(1)DNA extraction and PCR products’ amplification


Total microbial genomic DNA was extracted from soil samples using the E.Z.N.A.^®^ soil DNA Kit (Omega Bio-tek, Norcross, GA, USA) following the product manuals. Before being used further, the DNA was stored at −80 °C. Its quality and concentration were assessed using a 1.0% agarose gel electrophoresis and a NanoDrop2000 spectrophotometer (Thermo Scientific, Waltham, MA, USA). The hypervariable region V3-V4 of the bacterial *16S rRNA* gene was amplified with primer pairs 338F (5′-ACTCCTACGGGAGGCAGCAG-3′) and 806R (5′-GGACTACHVGGGTWTCTAAT-3′) [27] using a T100 Thermal Cycler PCR thermocycler (BIO-RAD, Hercules, CA, USA). The ITS1-1F-F (CTTGGTCATTTAGAGGAAGTAA) and ITS1-1F-R (GCTGCGTTCTTCATCGATG C) were used for a PCR of the fungi ITS. The PCR reaction mixture included 0.8 μL of each primer (5 μM), 4 μL 5 × Fast Pfu buffer, 2 μL 2.5 mM dNTPs, 10 ng of template DNA, 0.4 μL Fast Pfu polymerase, and ddH_2_O to a final volume of 20 µL. The following conditions were used for the PCR amplification cycle: a 3-min initial denaturation at 95 °C, 27 cycles of denaturing at 95 °C for 30 s, annealing at 55 °C for 30 s, extending at 72 °C for 45 s, a single extension at 72 °C for 10 min, and concluding at 4 °C. Following the manufacturer’s instructions, the PCR product was extracted from a 2% agarose gel and purified using the PCR Clean-Up Kit (YuHua, Shanghai, China) and quantified using Qubit 4.0 (Thermo Fisher Scientific, Waltham, MA, USA).


(2)Illumina PE300/PE250 sequencing


Using an Illumina PE300/PE250 platform (Illumina, San Diego, CA, USA) and standard techniques from Majorbio Bio-Pharm Technology Co. Ltd. (Shanghai, China), purified amplicons were pooled in equimolar quantities and paired-end sequenced.


(3)Amplicon sequence processing and analysis


Fastp (0.19.6) was utilized for quality filtering the resultant sequences, and FLASH (v1.2.11) was employed for their combination [28]. Subsequently, the DADA2 plugin in the Qiime2 pipeline [29] (version 2020.2) was utilized for the de-noising of the high-quality sequences, resulting in a single-nucleotide resolution based on error patterns within samples. These variants of amplicon sequences are commonly referred to as DADA2-denoised sequences (ASVs). To minimize the impact of sequencing depth on alpha and beta diversity measurements, the number of sequences from each sample was filtered to 20,000. This process yielded an average Good’s coverage of 97.90%. Taxonomic assignment of ASVs was carried out using the SILVA 16S rRNA database (v138), and a consensus taxonomy classifier built in Qiime2.

### 2.4. Data Processing and Analysis

SPSS and Origin were used to process and draw the plant-related and soil physical and chemical property data.

The Majorbio Cloud platform (https://cloud.majorbio.com, accessed on 1 March 2023) was used for bioinformatic analysis. To calculate rarefaction curves and alpha diversity indices based on the data from the ASVs, including observed ASVs, Chao1 richness, the Shannon index, and Good’s coverage, Mothur v1.30.1 [30] was utilized. Principal coordinate analysis (PCoA) based on the Bray–Curtis dissimilarity was conducted using the Vegan v2.5-3 software to assess the similarities between the microbial communities in various samples. Subsequently, the statistical significance of the variation explained by the treatment and its percentage was determined by conducting the PERMANOVA test using the Vegan v2.5-3 package. Additionally, LefSe [31] (http://huttenhower.sph.harvard.edu/LEfSe, accessed on 1 March 2023) was utilized to perform a linear discriminant analysis effect size (LDA > 2, *p* < 0.05) in order to identify bacterial taxa with substantial differences in phylum to genus abundance amongst various groups.

## 3. Results

### 3.1. Natural Recovery of Vegetation after Subsidence in Grassland Mining Areas

As shown in Figure 2, the species composition of the study area was mainly Asteraceae, legumes, and amaranth plants. Compared to one or two annual herbaceous species, there were substantially more perennial herbaceous species. The number of sample species after subsidence was greater than that of non-subsidence sample species; it first increased and then decreased with the increase in subsidence time. As the subsidence time increases, the vegetation cover and biomass recover year by year. There were significant differences in coverage and biomass between the non-subsidence and the subsidence plot after 1, 2, and 5 years, and there was no significant difference between the non-subsidence plot and the subsidence plot after 10 and 15 years. Compared with the non-subsidence group, the vegetation cover after 10 years of subsidence exceeded that of the non-subsidence group, but its biomass did not show the same regularity. Subsidence significantly reduced vegetation species diversity; both the H and E indexes increased first and then slowly decreased with the increase in subsidence time, indicating that the community gradually reached a relatively stable stage. The difference was that the H index had exceeded the non-subsidence group after 2 years of subsidence, but the E index in the subsidence group was lower than that of the non-subsidence group. This finding shows that, during the natural recovery process, it is difficult for the plant community to recover after subsidence to the same state it was in when it was not submerged.

### 3.2. Changes in Soil Physical–Chemical Properties after Subsidence in Grassland Mining Areas

As shown in Figure 3, the area was mainly composed of sand grains (62.90%) before being disturbed, with a fine sand content of 27.58%, a powder content of 4.15%, and a clay content of 5.37%. The coarse sand grain content decreased significantly due to subsidence; with the increase in subsidence time, the soil texture gradually changed from clay to sand, but it was sandy loam after 15 years of subsidence and did not recover to sand. The soil water content of most samples was between 15% and 30%; it exhibits a pattern of initially increasing before subsequently decreasing, reaching a maximum of 27.33% at 2 years of subsidence and then decreasing year by year, except for 10 years of subsidence with a soil water content of 60%. Soil water content after 10 years of subsidence was significantly different from other subsidence years and the non-subsidence soil. The soil pH in this area is mainly between 6 and 8 and is classified as neutral soil. Compared with non-subsidence soil (7.43), the soil pH was relatively stable after subsidence.

As shown in Figure 4, compared with the non-subsidence soil, the soil organic matter (SOM) content after settlement was significantly higher than that of the non-subsidence soil and then decreased roughly year by year, except for subsidence 2 years. There was little overall change in total soil total nitrogen (TN) and total phosphorus (TP) content, except for a significant decrease in total nitrogen and a significant increase in total phosphorus in the second year of subsidence. For total potassium, there was a significant increase in total potassium content in the soil after subsidence.

### 3.3. Changes in the Soil Microbial Community after Subsidence in Grassland Mining Areas

As presented in Table 1, subsidence will lead to an increase in the number of bacteria and fungi. However, the number of bacteria is lower than that in the non-subsidence group after 5 years of subsidence, and the number of fungi is greater than that in the non-subsidence group. The number of bacterial sequences in the early stage of subsidence was the largest, and then it showed a downward trend year by year. However, while the number of fungal sequences increased after subsidence, there was no obvious trend with the subsidence time.

As shown in Figure 5, subsidence is a form of negative feedback for bacteria; that is, subsidence weakens the activity intensity of bacterial communities and reduces their diversity as a whole. In contrast, subsidence is a form of positive feedback for fungi; that is, subsidence enhances the activity intensity of fungi and increases the diversity of fungal communities. The different subsidence times affected the community structure of soil bacteria and fungi, but the difference was not obvious. For bacterial communities, the similarity between bacterial communities at 15 years of subsidence and non-subsidence soil microbial communities was higher. Bacterial microbial communities with 1 year of subsidence were unstable, and their sample sites were relatively scattered; meanwhile, the soil bacterial communities of 2 and 5 years of subsidence were more similar, but the bacterial communities at 10 years of subsidence were significantly different and differentiated from those of other years to a certain extent, indicating that the bacterial community had a large change after 10 years of subsidence. However, bacterial communities showed no obvious trend with the increase in subsidence time. For fungal communities, the soil fungal community after subsidence was very different from the non-subsidence soil fungal community, and the fungal community after 15 years of subsidence was closer to the microbial communities of 1, 2, and 5 years of subsidence, indicating that with the advancement of natural recovery time, the fungal community did not change greatly.

As shown in Figure 6, for bacteria, at the phylum level, the dominant bacterial phylum are *Actinobacteria*, *Proteobacteria*, *Acidobacteria*, *Chloroflexi*, and *Firmicutes*, which together account for more than 90% of the bacterial population. Among them, *Actinobacteria* is the most dominant phylum, accounting for about 35%; *Proteobacteria* is the subdominant phylum, accounting for about 20%; *Acidobacteria* is slightly lower than *Proteobacteria* overall, accounting for about 15%; *Acidobacteria* is equal to or slightly higher than Proteobacteria in some subsidence year groups; the proportion of *Chloroflexi* fluctuates around 10%; and *Firmicutes* differed significantly in different groups, ranging from 2% to 20%. The *Actinobacteria* phylum showed the greatest changes during natural recovery after subsidence. In the first year just after subsidence, the *Actinobacteria* phylum increased dramatically with its proportion accounting for almost half of all bacteria. Subsequently, it recovered to almost the same level as it had before subsidence. In contrast, the *Firmicutes* phylum gradually decreases during the 1–2 years of subsidence, after which it gradually increases with natural recovery, ultimately returning to a level almost identical to that before the subsidence.

For fungi, at the phylum level, the dominant phyla are *Ascomycota*, *Mortierellomycota*, *Basidiomycota*, and *Glomeromycota*. *Ascomycetes* in all groups were above 67%, except for the soil 2 years after subsidence, where the percentage of *Ascomycetes* was only 53%. The percentage of *Ascomycota* after 15 years of natural restoration reached 78%, which is even higher than the level before subsidence. During the period of subsidence, the percentage of *Mortierellomycota* initially experienced a decrease in levels, followed by an increase, and ultimately returned to levels similar to those before the subsidence occurred. *Basidiomycota* accounts for an average of about 8% and did not change much during the subsidence and natural restoration process. *Glomeromycota* can account for up to 7% at 2 years of subsidence, but it is almost impossible to find after 15 years of natural recovery.

### 3.4. Key Factors in the Natural Recovery of the Soil Microbial Community

The results of the RDA analysis of 14 environmental factors, namely, biomass, species number (SN), coverage, E, SW, pH, mechanical composition, coarse sand content, and soil chemical properties (i.e., organic matter, total nitrogen, nitrate nitrogen, ammonium nitrogen, available phosphorus, total potassium, and available potassium), are shown in Figure 7. For bacteria, the explanatory degrees of the first and second axes of RDA were 27.54% and 22.08%, respectively. The first axis was mainly composed of SW, pH, SOM, NH_4_-N, TK, AK, E, coverage, and biomass, and the second axis mainly comprised fine sand content and AP. For fungi, the explanatory degrees of the first and second axes of the RDA were 47.75% and 7.73%, respectively. The first axis was mainly composed of SOM, TK, and AK, and the second axis was mainly composed of SW, E, SN, and biomass.

The SN, coverage, TN, NH_4_-N, fine sand content (fine), and AP had a large impact on bacterial samples, among which AP was positively correlated with fine sand content, NH_4_N was positively correlated with TN, SN was positively correlated with coverage, and AP/fine sand content had no correlation with SN/NH_4_N and was negatively correlated with coverage/TN. The influence of all environmental factors on fungi was more significant. The sample points with different subsidence times were scattered and did not have obvious characteristics with environmental factors.

Soil microorganisms were analyzed for the Spearman correlation between species and environmental factors at the phylum classification level, and correlation heat maps were obtained. As shown in Figure 8, the dominant bacterial phylum, *Actinobacteria*, was significantly positively correlated with TN, very positively correlated with NO_3_-N and NH_4_-N, and negatively correlated with the number of species. There was no obvious correlation between *Proteobacteria* and environmental factors; *Acidobacteria* was significantly positively correlated with biomass; and *Chloroflexi* was significantly positively correlated with NO_3_-N and negatively correlated with AK and species number. There was no obvious correlation between *Firmicutes* and environmental factors. There was no significant correlation between *Firmicutes* and environmental factors in the phylum, with significantly different phylum abundances observed between different groups. *Gemmatimonadetes* were significantly correlated with several environmental factors, such as being highly positively correlated with SW, SOM, TK, and AP and positively correlated with AK and SN. They were negatively correlated with fine mechanical composition, and E. *Desulfobacterota* was significantly negatively correlated with SOM and AK, very negatively correlated with TK, significantly positively correlated with NH_4_-N, and highly positively correlated with biomass. Strong positive correlations were found between *Bdellovibrionota* and NH_4_-N and strong negative correlations were found between RCP2-54 and TK and AK. There was a substantial negative correlation found with coverage and biomass and a significant positive correlation with SOM, TK, and AK for *Deinococcota*. Significantly adversely connected with SOM, TK, AK, and AP, extremely negatively correlated with NH_4-_N and NO_3-_N, and significantly positively linked with E and biomass were the relationships observed between *Spirochaetota* and these variables. *Margulisbacteria* showed a strong positive correlation with NO_3_-N and NH_4_-N and a substantial negative correlation with SW, TK, and AP.

As shown in Figure 9, the dominant phylum of fungi, *Ascomycota*, was significantly positively correlated with TK and AK. *Unclassified_k_Fungi* had no obvious correlation with environmental factors. *Mortierellomycota* was positively correlated with SW, AP, and SN. *Basidiomycota* and *Glomeromycota* were not significantly correlated with environmental factors. *Mortierellomycota* was significantly positively correlated with SW, AP, and SN among the phyla with significantly different phylum abundance. There was a substantial negative correlation between *Rozellomycota* and SOM, while there was a positive correlation between AK and TK and plant biomass and coverage. *Aphelidiomycota* exhibited a substantial negative correlation with NH_4_-N, while demonstrating positive correlations with TK, AK, and SN, with AK showing the most positive correlation.

## 4. Discussion

### 4.1. The Regulation of the Natural Restoration of Vegetation

Vegetation coverage, biomass, and diversity decreased significantly in the early stage of subsidence. These factors gradually increased with the increase in subsidence years, with coverage and diversity basically returning to the pre-subsidence level. However, there was still a certain gap between biomass before and after subsidence, which was consistent with the results of Zhang et al. [32] on vegetation restoration in different subsidence areas. From the perspective of species population, the number of species gradually increased after subsidence, reached a maximum value at 5 years of subsidence, and then gradually decreased. This may be related to the degradation of grasslands caused by subsidence, resulting in the spread of a large number of weeds. The number of species in the subsidence period of 1 year was the lowest, indicating that the grassland was seriously degraded and biodiversity was seriously lost [33]. With the increase in sedimentation time, the dominant species changed from one or two annual herbs to perennial herbs and from single life forms to multiple life forms. Perennial plants have a stronger ability to resist environmental disturbances and maintain population stability than annual plants, so this change in the composition of species also reflects the changes in ecosystem structure and functioning during the process of vegetation restoration; the community structure tends to be stable, and the ecological function is enhanced [34]. However, in general, although natural restoration can partially restore aboveground vegetation, it is difficult to restore the area to its pre-subsidence state; rather, it forms a new homeostasis that is suitable for the environment, which is consistent with the conclusion of Liu et al. [35] that vegetation of a coal mining subsidence area can only be restored to a state close to its original state under natural restoration state.

### 4.2. The Natural Restoration of Soil Physical and Chemical Properties Changes Regulation

Although it has been reported that coal mining causes a decrease in soil water content [36], subsidence instead causes soil water content to be somewhat higher than before subsidence due to the high groundwater table in this study area. This is especially more pronounced in group year 10, but it may be an individual case. As the method of replacing time with space was used in this study when selecting representative areas at different stages of subsidence, it was very difficult to keep all the conditions consistent in field experiments. Subsidence also leads to changes in the physical structure and chemical form of soil, and the proportion of fine particles with a diameter of less than 0.2 mm in the soil increases significantly after subsidence, thereby increasing the soil’s water-holding capacity. At the same time, subsidence also caused the organic matter content, total nitrogen, total phosphorus, and total potassium in the soil to rise significantly after subsidence and then gradually decrease, which is consistent with the conclusion of Wu et al.’s [37] research in a coal mining subsidence area of the semi-arid region of Northwest China. This may be because the cracks and slopes caused by subsidence will promote the movement and loss of surface materials. The slope formed by subsidence will cause the gradual downward transfer of nutrients and fine particles in the soil [10], resulting in a high content of fine particles and nutrients in the soil at the beginning of subsidence that, over time, will be transported alluvially to the bottom of the slope or waterlogged areas. The content of fine particles and nutrients will tend to be stable. Song et al. [38] found a similar phenomenon in their study of subsidence soils in coal mining areas in northern Shaanxi.

Relevant studies have shown [39] that the proportion of soil organic carbon and soil clay particles is significantly positively correlated, and the higher the proportion of clay particles, the stronger the stability of soil organic carbon in the soil. In this study, the ratio of soil organic matter to clay particles also showed the same pattern. At the same time, studies have shown [40] that soil organic carbon is mainly regulated by soil pH and plant biomass. Total soil nitrogen first increased and then decreased with the increase in time, nitrate nitrogen content decreased first and then increased, and ammonium nitrogen content gradually decreased with the increase in subsidence time and then flattened. The results of this study also confirm the positive correlation between soil total nitrogen and soil clay particles [38]. However, with the increase in subsidence time, there was no obvious change trend in soil total phosphorus, but the available phosphorus generally increased. Studies have shown that different phosphorus application rates significantly affect plant photosynthetic characteristics and ultimately affect plant biomass accumulation, but excessive phosphorus content can also produce inhibitory effects [41].

### 4.3. Changes in Soil Microbial Communities in Natural Restoration

Soil microorganisms are sensitive to environmental changes caused by soil subsidence [42,43], and in this study, it was found that soil subsidence significantly impacts the structure and diversity of microbial communities. Although the number of bacterial species and community diversity recovered over time, these metrics were far lower than those in the non-subsidence areas, which may be because subsidence from coal mining altered the physical and chemical characteristics of the soil, which in turn influenced bacterial colonization in the disturbed areas and decreased the diversity of bacterial communities [44,45]. However, for fungi, subsidence increased the number of fungal species and community diversity. Specifically, the number of species decreased in the later stage, but this decrease was not obvious. The diversity of fungal communities increased rapidly in the first 5 years and then decreased significantly. These results indicate that bacteria and fungi are sensitive to changes in the soil environment caused by subsidence, but there are obvious differences in the reactions between the two, which are also related to the different environmental needs of fungi and bacteria. Bacteria and fungi have different nutritional preferences [46], with bacteria preferring to use easily decomposing carbon sources, which are more susceptible in the early stages of disturbance, while fungi can use carbon sources that are difficult for bacteria to decompose, so the effect of disturbing fungi is not very large [47].

From the perspective of community structure, subsidence also had a certain impact on the community composition of soil bacteria and fungi. Many factors affect the structure of microbial communities, such as environmental and vegetation conditions [48]. In this study, although subsidence led to changes in soil physicochemical properties and vegetation conditions, there was no significant change in the soil microbial community’s structure after subsidence and during natural restoration. This is consistent with the community composition results; from the phylum level, whether it is fungi or bacteria, the dominant bacteria do not change at different time stages of natural recovery before and after subsidence. However, the proportions of different dominant bacteria in the whole community are slightly different at different stages of subsidence, which is also a result of the influence of environmental and vegetation conditions during the subsidence process [49]. As one of the most widely distributed bacterial phyla in soil, *Actinobacteria* showed a sudden increase in their percentage in the first year just after subsidence in this study, which may be related to the key ecophysiological role of *Actinobacteria* in decomposing plant residues [50]. Aboveground biomass and cover of vegetation in the first year of subsidence were remarkably reduced compared to the pre-subsidence period, suggesting that there was significant vegetation mortality. The residues of the dead vegetation provided abundant food for the *Actinobacteria*, leading to their proliferation. As natural recovery occurred, the vegetation cover and biomass gradually recovered, and the number of *Actinobacteria* gradually returned to the level before subsidence. Our study also found, during the first 2 years after the subsidence, a gradual decrease in the *Firmicutes* phyla with the degradation of the grassland, which is in agreement with the results of the study in the temperate grassland. With natural recovery, the thick-walled phylum gradually recovered to the pre-sinking level. The largest percentage of fungi are *Ascomycetes*, known for their preference for challenging environments and significant involvement in the decomposition of complex organic matter [51].

### 4.4. The Role of Soil Microbial Communities in Natural Restoration

In this study, the number of aboveground vegetation species, coverage, fine sand content in the soil, TN, NH_4_N, and AP are the dominant factors influencing the bacterial community as can be observed from the results of the RDA analysis and heatmap. Sun et al. [52] also showed that aboveground vegetation had a significant impact on the structure of bacterial communities in soil and pointed out that SOM had the greatest impact on the composition and distribution of bacterial communities, and AP played an important role in the distribution of bacteria. Compared with previous studies, they showed that soil total carbon, AP, and AN strongly promoted soil bacterial community diversity. However, there was little correlation between TN content and bacterial community [53], indicating that the bacterial community structure in soil was related to different nitrogen contents. The results showed that the pH value was the dominant factor in the horizontal structure of bacterial communities in the coal mining subsidence area. However, in this study, the pH change was not obvious, and the relationship with bacterial diversity was not significant, which matched the results of Du et al. [18].

However, there are many dominant factors affecting the fungal community, and almost all the factors investigated, including soil physicochemical properties and vegetation conditions, will have a significant impact on the composition of the fungal community, which is also the reason why the fungal community structure changes greatly after subsidence. Compared with bacteria, aboveground vegetation is more pronounced for fungi, which may be because as vegetation biomass increases, vegetation provides resource heterogeneity to soil microbial communities through various factors such as litter decomposition and root exudates, so higher biomass leads to better coexistence of microbial communities [54]. Similarly, fungi are more sensitive to changes in the soil environment. The dominant fungal phyla include *Ascomycetes*, *Basidiomycetes*, and *Mycobacteria*, similar to the findings of other researchers [55]. Among these, *Ascomycetes* and *Basidiomycetes* are considered to be the main fungal decomposers in soils due to their broad-spectrum degradation capacity [56]. *Ascomycetes* dominate soils globally due to their ability to adapt to a variety of environments [57,58], and they are saprophytic fungi that play an important role in soil material cycling. However, *Basidiomycetes* can degrade lignin and other recalcitrant organic substances [59], which may be the reason for the relative abundance of *Basidiomycetes* in grasslands. Therefore, soil organic matter plays an important driving role in fungal community composition.

## 5. Conclusions

Coal mining subsidence altered the soil’s microbiological community, physical and chemical characteristics, and aboveground vegetation community in the affected area. In the process of 15 years of natural restoration, although the microbial communities in the aboveground vegetation and soil were partially restored, it is difficult for them to return to their pre-subsidence state. However, a new homeostasis adapted to the environment was formed. Soil physicochemical properties and vegetation conditions were found to have a significant impact on the composition of bacterial and fungal communities, especially the number of aboveground vegetation species, vegetation coverage, and the content of fine sand particles, TN, NH_4_-N, and AP in the soil, which is also an important indicator for assessing the effectiveness of ecological restoration. This study revealed the response of the soil microbial community to the natural restoration of subsidence by studying the changes and influencing factors of microbial diversity and community structure in different natural restoration processes, which has important guiding significance for the efficient ecological restoration of coal mining subsidence areas.

## Figures and Tables

**Figure 1 microorganisms-12-00087-f001:**
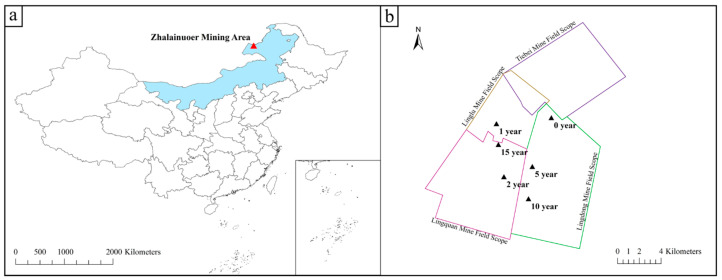
(**a**,**b**) Schematic map of the study area.

**Figure 2 microorganisms-12-00087-f002:**
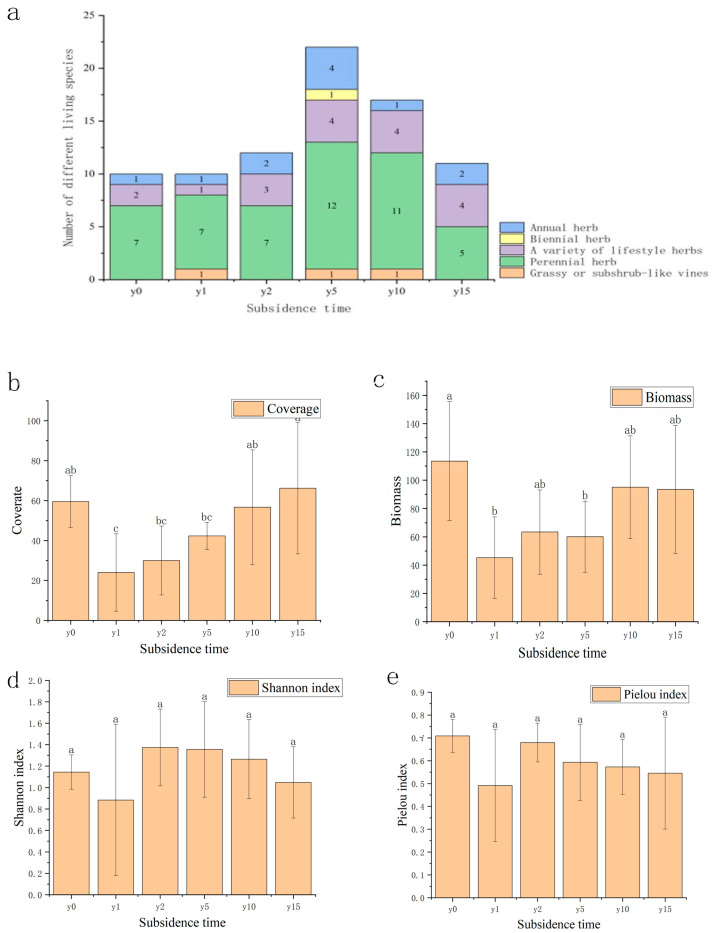
(**a**) Variation diagram of species quantity and subsidence time of different lifeforms. (**b**) Variation diagram of plant coverage and subsidence time. (**c**) Variation diagram of biomass and subsidence time. (**d**) Variation diagram of Shannon index under different subsidence times. (**e**) Variation diagram of Pielou index under different subsidence times. Different letters indicate significant differences.

**Figure 3 microorganisms-12-00087-f003:**
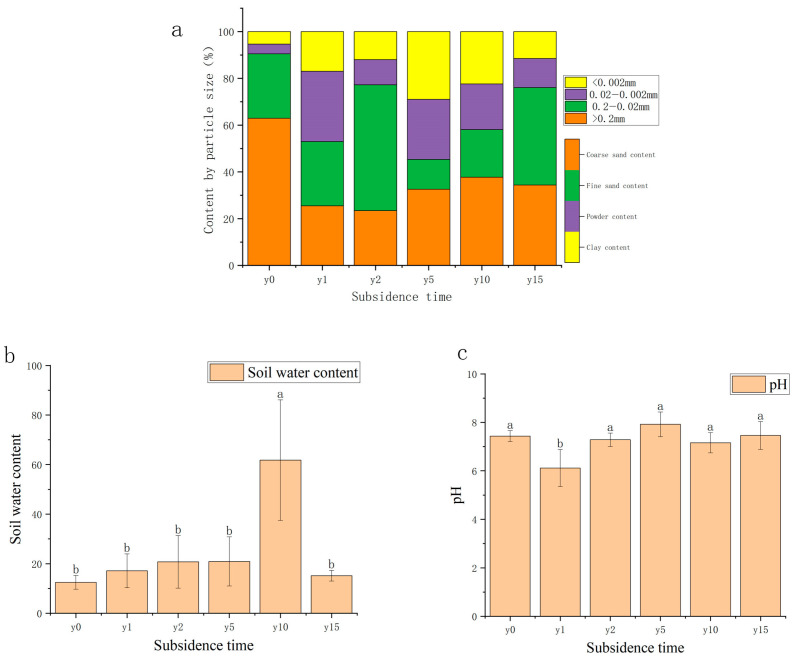
(**a**) Variation diagram of mechanical composition and subsidence time. (**b**) Variation diagram of water content and subsidence time. (**c**) Variation diagram of soil pH and settlement time. Different letters indicate significant differences.

**Figure 4 microorganisms-12-00087-f004:**
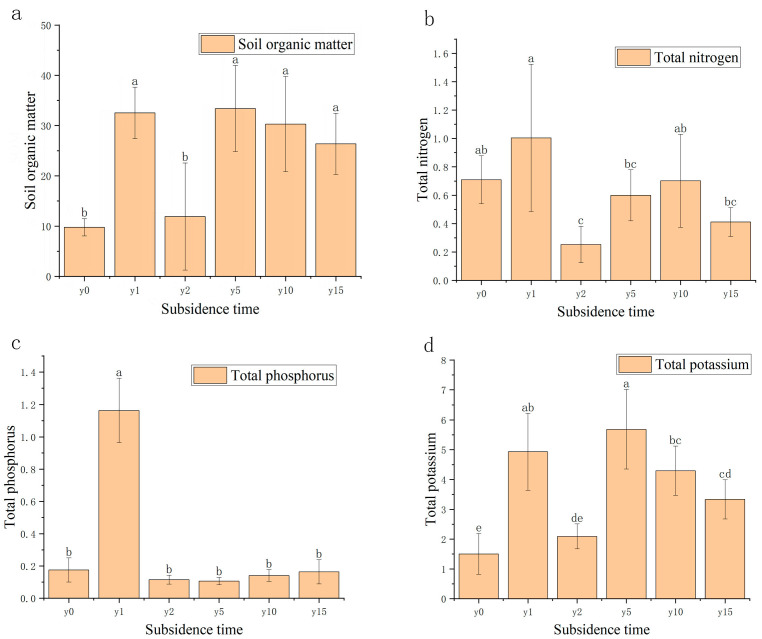
(**a**) Variation diagram of soil organic matter and subsidence time. (**b**) Relationship between soil total nitrogen, nitrate nitrogen, and ammonium nitrogen and subsidence time. (**c**) Relationship between soil total phosphorus, available phosphorus, and subsidence time. (**d**) Relationship between soil total potassium, available potassium, and subsidence time. Different letters indicate significant differences.

**Figure 5 microorganisms-12-00087-f005:**
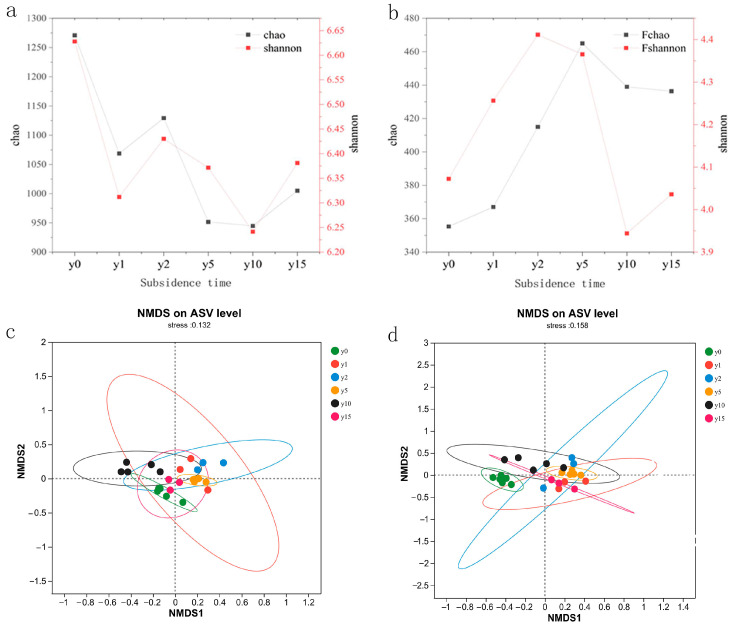
(**a**) Bacterial alpha diversity index of different subsidence times. (**b**) Fungal alpha diversity index of different subsidence times. (**c**) NMDS analysis of soil bacteria at different subsidence times. (**d**) NMDS analysis of soil fungi at different subsidence times.

**Figure 6 microorganisms-12-00087-f006:**
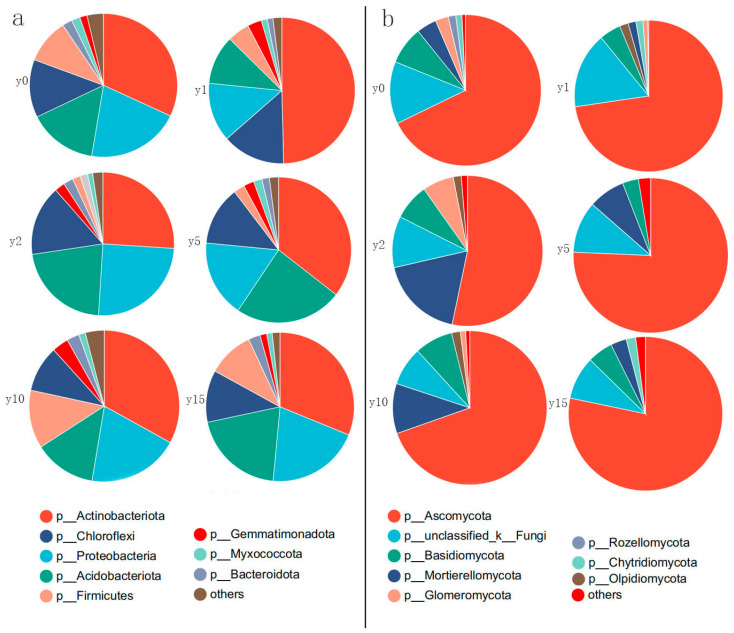
Relative abundance of bacterial phyla (**a**) and fungal phyla (**b**) at different subsidence times.

**Figure 7 microorganisms-12-00087-f007:**
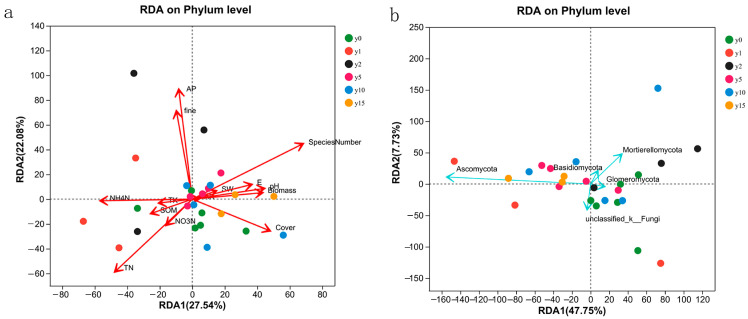
RDA analysis of bacteria (**a**) and fungi (**b**) samples and environmental factors at different subsidence times.

**Figure 8 microorganisms-12-00087-f008:**
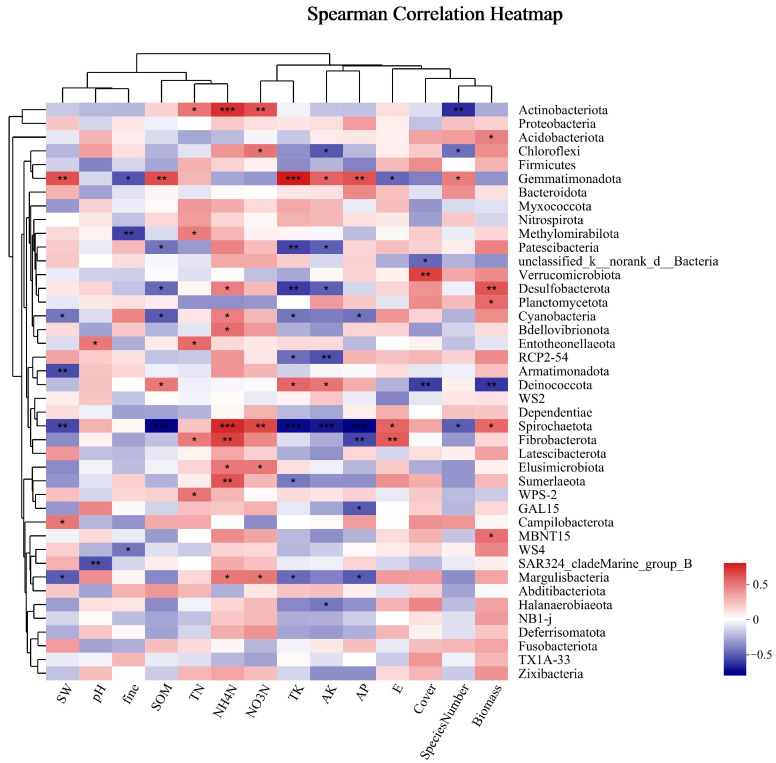
Cluster thermogram of horizontal abundance of soil bacteria microbial community and environmental factors. *: The correlation is significant at the level of 0.05; **: The correlation is significant at the level of 0.01; ***: The correlation is significant at the level of 0.001.

**Figure 9 microorganisms-12-00087-f009:**
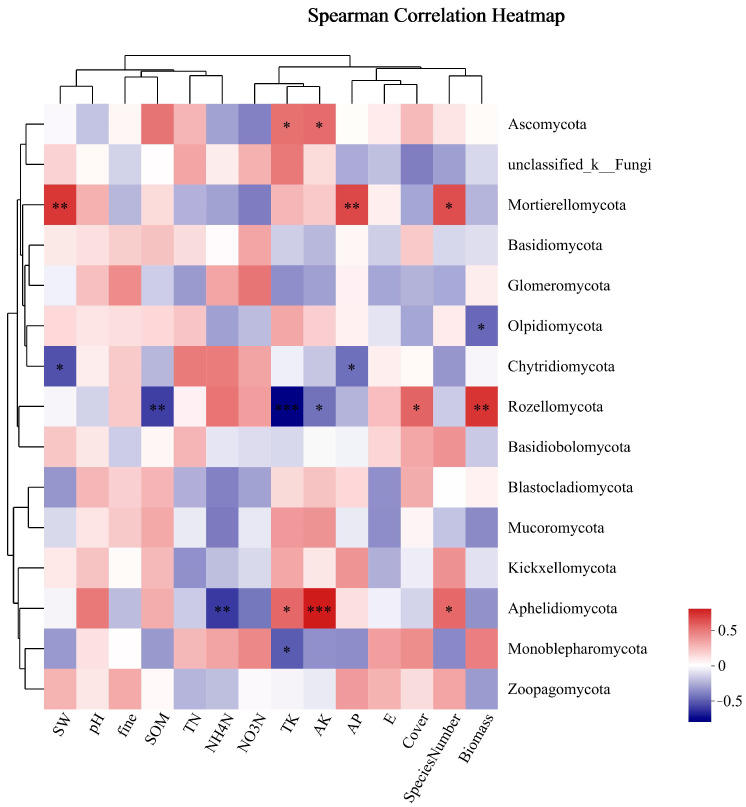
Cluster thermogram of horizontal abundance of soil fungal microbial community and environmental factors. *: The correlation is significant at the level of 0.05; **: The correlation is significant at the level of 0.01; ***: The correlation is significant at the level of 0.001.

**Table 1 microorganisms-12-00087-t001:** ASV quantity in high-throughput sequencing.

Constituencies	y0	y1	y2	y5	y10	y15
Bacteria	25567 ± 1677 ^ab^	27869 ± 5611 ^a^	25807 ± 3405 ^ab^	22675 ± 2630 ^ab^	24452 ± 1858 ^ab^	22298 ± 3293 ^b^
Fungi	38866 ± 1682 ^b^	53242 ± 4685 ^a^	40946 ± 5386 ^b^	45588 ± 7322 ^ab^	43380 ± 6523 ^ab^	49403 ± 4410 ^a^

Note: a,b: in the same column, values with different superscript letters differed significantly (*p* < 0.05).

## Data Availability

Data is contained within the article. The data presented in this study are available in this manuscript.

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
