# Peer review of "Driving Factors Influencing Soil Microbial Community Succession of Coal Mining Subsidence Areas during Natural Recovery in Inner Mongolia Grasslands"

_microorganisms, 2023, doi:10.3390/microorganisms12010087_

Round 1

Reviewer 1 Report (Previous Reviewer 1)

Comments and Suggestions for Authors

This study investigates the natural restoration progression in a mining area over periods of 1, 2, 5, 10, and 15 years, exploring its effects on vegetation, soil, and the microbiome. Despite its genuine significance and comprehensive nature, the article falls short in effectively communicating the conducted research and requires substantial revisions before undergoing a thorough review.

Drawing on my professional experience, my primary focus lies in the microbiological analysis. Both the results and the discussion pertaining to the analysis of soil microbiological diversity are notably limited. Little attention is given to variations among different phyla, families, orders, and genera, as well as their temporal evolution. While differences in microbial group recovery rates might exist, this aspect is barely addressed. Merely a paragraph is dedicated to discussing Figure 6, and the associated discussion in Section 4.3 appears generic and lacking substance.

Within the results of Section 3, certain samples deviate significantly from the values of the remaining samples (e.g., y10 in Figure 3b, y2 in Figure 4A, or y1 in Figure 4C, among others), yet the reason for these variations is not explained during the course of the discussion.

In essence, it is a commendable piece of work, but its potential has not been fully realized.

Minor Revisions:

Review the English language throughout the entire paper (e.g., line 16: "as well as, and"; lines 49-54; line 151: "iincluded").

Alphabetize the keywords.

Line 135: Are all the protocols described in reference 24?

Italicize "16S rRNA" as it is a gene.

Review subscripts and superscripts (e.g., see ddH2O, or NH4N).

Indicate groups with significant differences using letters in the tables.

Comments on the Quality of English Language

Moderate editing of English language required

Author Response

Dear Editors and Reviewers:

We are truly grateful for your and other reviewers’ valuable comments and beneficial suggestions concerning our manuscript (microorganisms-2781964). Your comments and those of the reviewers were highly insightful and enabled us to greatly improve the quality of our manuscript. In the following pages are our point-by-point responses to each of the comments of the reviewers as well as your comments. Revised portions are highlighted in yellow. We hope that the revisions in the manuscript and our accompanying responses will be sufficient to make our manuscript suitable for publication.

We shall look forward to hearing from you at your earliest convenience.

Kind regards,

Zhen Mao

Ph.D, Associate Professor

School of Environment Science and Spatial Informatics, China University of Mining and Technology, Xuzhou 221116, Jiangsu Province, P. R. China

Responds to the reviewers comments:

Reviewer #1:

  1. Comment: Drawing on my professional experience, my primary focus lies in the microbiological analysis. Both the results and the discussion pertaining to the analysis of soil microbiological diversity are notably limited. Little attention is given to variations among different phyla, families, orders, and genera, as well as their temporal evolution. While differences in microbial group recovery rates might exist, this aspect is barely addressed. Merely a paragraph is dedicated to discussing Figure 6, and the associated discussion in Section 4.3 appears generic and lacking substance.

Response: Thank you for your suggestion. We have enriched the results and discussion of the microbiological analyses (Line 286-307 and Line 468-483) and added some references accordingly.

  1. Comment: Within the results of Section 3, certain samples deviate significantly from the values of the remaining samples (e.g., y10 in Figure 3b, y2 in Figure 4A, or y1 in Figure 4C, among others), yet the reason for these variations is not explained during the course of the discussion.

Response: Thank you very much for your suggestion. Although the study area was selected to be as representative as possible while expanding the sample size, individual data may vary by a wide margin due to the consistency of the soils and uncontrolled field conditions, but the overall trend remains unchanged. We discuss that in Line 397-403.

  1. Comment: Review the English language throughout the entire paper (e.g., line 16: "as well as, and"; lines 49-54; line 151: "iincluded").

Response: Thanks for your kind reminder. We corrected some minor errors during the revision process. All these changes have been highlighted in yellow.

  1. Comment: Alphabetize the keywords.

Response: Thanks for your suggestion. The keywords have been rearranged in alphabetical order

  1. Comment: Line 135: Are all the protocols described in reference 24?

Response: Only Total phosphorus (TP) and total potassium (TK) were tested according to the method of Reference 24.

  1. Comment: Italicize "16S rRNA" as it is a gene.

Response: Thanks for your suggestion. "16S rRNA" has been italicized.

  1. Comment: Review subscripts and superscripts (e.g., see ddH2O, or NH4N).

Response: We apologize for our carelessness. These errors have been corrected during the revision process.

  1. Comment: Indicate groups with significant differences using letters in the tables.

Response: The data was tested for differences among sites with one-way analysis of variance (ANOVA). And the results were shown in Table 1.

Reviewer 2 Report (Previous Reviewer 2)

Comments and Suggestions for Authors

The manuscript was substantially improved compared to its previous version. However, there are still some observations. 

Comments on the Quality of English Language

Some terms should not be used, such as "microflora" in the conclusions. 

The fluidity of language remains a problem; there is an overuse of the word "subsidence", which could be replaced by demonstrative connectors and pronouns.

The sections "Soil Microbial Samples" and "Data Processing and Analysis" must be completely paraphrased to avoid significant overlap with other sources.

Author Response

Dear Editors and Reviewers:

We are truly grateful for your and other reviewers’ valuable comments and beneficial suggestions concerning our manuscript (microorganisms-2781964). Your comments and those of the reviewers were highly insightful and enabled us to greatly improve the quality of our manuscript. In the following pages are our point-by-point responses to each of the comments of the reviewers as well as your comments. Revised portions are highlighted in yellow. We hope that the revisions in the manuscript and our accompanying responses will be sufficient to make our manuscript suitable for publication.

We shall look forward to hearing from you at your earliest convenience.

Kind regards,

Zhen Mao

Ph.D, Associate Professor

School of Environment Science and Spatial Informatics, China University of Mining and Technology, Xuzhou 221116, Jiangsu Province, P. R. China

Reviewer #2:

  1. Comment: Some terms should not be used, such as "microflora" in the conclusions. 

Response: Thank you for pointing this out. The word "microflora" has been changed in the new version.

  1. Comment: The fluidity of language remains a problem; there is an overuse of the word "subsidence", which could be replaced by demonstrative connectors and pronouns.

Response:  Thank you for your kind advice. However, “subsidence” is a technical term in the mining industry, and I'm very sorry that I didn't find a word that was too suitable to replace it, but we've tried to minimize the frequency of its use in the text.

  1. Comment: The sections "Soil Microbial Samples" and "Data Processing and Analysis" must be completely paraphrased to avoid significant overlap with other sources.

Response: Based on your suggestion, the sections "Soil Microbial Samples" and "Data Processing and Analysis" have been reorganized (Line 148-194).

Reviewer 3 Report (New Reviewer)

Comments and Suggestions for Authors

The manuscript entitled “Driving factors influencing soil microbial community succession of coal mining subsidence areas during natural recovery in Inner Mongolia grasslandsis aims to soil microbial communities succession in the natural recovery process after mining subsidence. The natural recovery mechanism of soil microorganisms was analyzed along with the changes related to vegetation and soil physicochemical properties.  This topic original and relevant in the field of soil science ecology and microbiology. The authors in this manuscript provided interdisciplinary data on soils for 15 years, including a description of the composition of microorganisms, changes in mechanical composition and physico-chemical properties. To my mind this manuscript is topical and corresponding to the aims and scopes of the “Microorganisms journal. The key results presented in the manuscript, in my opinion, show important aspects of soil-bacteria and fungi consortia evolution; the methods used in the work allowed the authors to obtain reliable results. The references used in the review and the text itself are relevant

in general, I liked the work, it is well thought out and logically constructed, written in simple, understandable language.

Here are the comments I found while reading the manuscript.

The abstract should contain more specific results obtained by the authors

112 указать при какой температуре

indicate which primers were used for the analysis of fungi

What was the thickness of the newly formed soil layer? Is it the same everywhere? samples were taken from a depth of 10 cm, what justified the choice of depth?

143 PCR products amplification

In conclusion, it is worth describing in more detail a large number of important results obtained by the authors, including indicating which of the selected aspects are more important in assessing the effectiveness of soil remediation

Author Response

Dear Editors and Reviewers:

We are truly grateful for your and other reviewers’ valuable comments and beneficial suggestions concerning our manuscript (microorganisms-2781964). Your comments and those of the reviewers were highly insightful and enabled us to greatly improve the quality of our manuscript. In the following pages are our point-by-point responses to each of the comments of the reviewers as well as your comments. Revised portions are highlighted in yellow. We hope that the revisions in the manuscript and our accompanying responses will be sufficient to make our manuscript suitable for publication.

We shall look forward to hearing from you at your earliest convenience.

Kind regards,

Zhen Mao

Ph.D, Associate Professor

School of Environment Science and Spatial Informatics, China University of Mining and Technology, Xuzhou 221116, Jiangsu Province, P. R. China

Reviewer #3:

  1. Comment: The abstract should contain more specific results obtained by the authors.

Response: Thank you for your kind advice. We've added a few sentences to the abstract to make it more specific (Line 17-20).

  1. Comment: 112 ÑƒÐºÐ°Ð·Ð°Ñ‚ÑŒ Ð¿Ñ€Ð¸ ÐºÐ°ÐºÐ¾Ð¹ Ñ‚емпературе indicate which primers were used for the analysis of fungi

Response: We presume our suggestion. The primers for PCR of fungi were given in Line155-156.

  1. Comment: What was the thickness of the newly formed soil layer? Is it the same everywhere? samples were taken from a depth of 10 cm, what justified the choice of depth?

Response: Thank you for pointing this out. The thickness of the new soil layer was only a few millimeters thick at the thickest of all the sampling sites. Therefore, we set the sampling depth at 10 centimeters to fully include the new soil layer.

  1. Comment: In conclusion, it is worth describing in more detail a large number of important results obtained by the authors, including indicating which of the selected aspects are more important in assessing the effectiveness of soil remediation

Response: Thank you for your suggestion. We point it out in Line 526-527.

Round 2

Reviewer 1 Report (Previous Reviewer 1)

Comments and Suggestions for Authors

The article has been properly corrected

Reviewer 3 Report (New Reviewer)

Comments and Suggestions for Authors

The authors have significantly improved the manuscript. In this form I am ready to recommend it for publication

This manuscript is a resubmission of an earlier submission. The following is a list of the peer review reports and author responses from that submission.

Round 1

Reviewer 1 Report

Comments and Suggestions for Authors

This study examines the evolution of natural restoration in a mining area over 1, 2, 5, 10, and 15 years and its impact on vegetation, soil, and the microbiome. Despite being a topic of real interest and a very comprehensive piece of work, the article falls short in effectively conveying the research conducted and requires major revisions before it can be properly reviewed.

To begin with, a thorough review of the writing and formatting is necessary, addressing basic issues such as proper separation between words and references, and the inclusion of periods between sentences (line 34, "Civilization [1,2]. China…," needs to be reviewed throughout the entire text).

The discussion section primarily reiterates the study's main results without substantial analysis. The paper contains only 37 references, of which 10 are related to the discussion. It lacks explanations for the results, and it is unclear whether they are significant or not because it doesn't convey the importance or magnitude of the findings.

The graphs should indicate significant differences between the sample groups using different letters. Throughout the study, it mentions high or small differences or similarities when it should treat them as significant or not.

The materials and methods section is concise and lacks references or protocols that would enable replication in another laboratory (see soil sample analysis).

Minor revisions:

Separate units from numbers (e.g., line 226: 5393.00 mg/kg, line 146: 10 cm).

Duplicate words (line 37: heatmap, line 312: biomass), expressions (line 338: species number, line 342: positively correlated with AK), and even entire paragraphs (lines 86 to 95) are common.

I have no doubt about the work's quality, but I believe that the presentation of the study does not allow for an assessment of its merit. I recommend a comprehensive review of the written work and submitting it when it is complete and error-free.

Comments on the Quality of English Language

The whole text should be revised

Reviewer 2 Report

Comments and Suggestions for Authors

The manuscript Driving factors influencing soil microbial community succession of coal mining subsidence areas during natural recovery in Inner Mongolia grasslands, is a well research. 

However, there are many details that need to be included in the manuscript. 

There are several observations regarding the grammar. 

For example: repetition of "years" in the abstract L17; misplacing of a comma L24; redundance of the word "China" L34. Redundancy of "area" L86-87. Inconsistences as "taken back" to where? (L148). Incorrect verbal times (L129 – were instead of are). Material and methods subheadings are not informative, e.g., "Soil microbial samples" to describe soil microbial community characterization. 

Almost all paragraphs have missing spaces after the point (L31-70). 

Other specific observations are:

References are missing in the site climatic description (L104-113). Also, check the units' presentation. Specify primary vegetation (previous and current) using scientific names. 

Distance between soil sampling points must be specified. Provide a reference for SOM protocol determination.

If the authors use preestablished primers (i.e., ITS), this must be specified. Otherwise, the deposit of genetic information must be provided. Condition for DNA extraction and amplification; references of all protocols must be provided. 

Excel does not have the scientific precision to build scientific graphs.

Statistical analysis procedure details must be provided in the "Analysis" section. 

Many parts of the discussion do not make any comparison with the existing literature. So there are many missing references in this section. L364-374 - is a clear example of the deficiency in the writing and the lack of citations of works that support the statements. Except for the last idea, it is just a description of the results. This is just an example, the entire discussion must be restructured to improve the transmission of the knowledge produced in this research.

No reference is provided for plant-soil microbial feedback taxa. L449-491 – there are few and, in some cases, non-reference to the applicability of this feedback to the study. This section seems more like an introduction. A clear connection between the theory and the founding of the study must be specified. 

The writing of the conclusions is very poor and not very correct. At what time and under what criteria were it specified that "macroscopic performance" referred to "vegetation cover and diversity index, soil moisture content and pH." There are more scientific terms to refer to these characteristics of the vegetation (cover and diversity) and the physical and chemical properties of the soil. The same applies to "internal composition".

"nitrogen-related indexes" are not the same as total nitrogen, nitrate nitrogen, and ammonium nitrogen content; the same for "potassium-related indexes". Review the definition of "index".

The authors did not test "microbial evolution and microbiota function," so they cannot conclude anything about it.

Comments on the Quality of English Language

English is very poor throughout the manuscript, and it is difficult to evaluate the results and discussion as a result of these language deficiencies. The manuscript must first be improved in this aspect to assess scientific robustness.